# Deep-subwavelength control of acoustic waves in an ultra-compact metasurface lens

Jian Chen[1], Jing Xiao[1], Danylo Lisevych[1], Amir Shakouri[1] & Zheng Fan [1]

Space-coiling acoustic metasurfaces have been largely exploited and shown their outstanding wave manipulation capacity. However, they are complex in realization and cannot directly manipulate acoustic near-fields by controlling the effective path length. Here, we propose a comprehensive paradigm for acoustic metasurfaces to extend the wave manipulations to both far- and near-fields and markedly reduce the implementation complexity with a simple structure, which consists of an array of deep-subwavelength-spaced slits perforated in a thin plate. A semi-analytical approach for such a design is established using a microscopic coupled-wave model, which reveals that the acoustic diffractive pattern at every slit exit is the sum of the initial transmission and the secondary scatterings of the coupled fields from other slits. For proof-of-concept, we examine two metasurface lenses for sound focusing within and beyond the diffraction limit. This work provides a feasible strategy for creating ultra-compact acoustic components with versatile potentials.

[1] School of Mechanical and Aerospace Engineering, Nanyang Technological University, 50 Nanyang Avenue, Singapore 639798, Singapore. Correspondence and requests for materials should be addressed to Z.F. (email: ZFAN@ntu.edu.sg)

The convergence of acoustic energy is crucial in a wide variety of applications, including ultrasound imaging and therapy, wave energy harvesting, acoustic communication, and particle manipulation. Acoustic refractive lenses are widely used for focusing acoustic waves[1]. In general, they rely on wave propagation over distances that are much larger than the wavelength. In this way, phase changes can be engineered by controlling the acoustic path lengths, either by curving the interfaces or by varying the refractive index throughout the entire volume of the lens[2]. Although acoustic refractive lenses can focus acoustic waves very effectively, they are hindered by large dimensions, low throughput, and massive material costs because of the acoustic properties of natural materials. Acoustic Fresnel lenses provide an alternative approach for focusing by alternating rigid (opaque) and open (transparent) apertures for constructive interferences. Fresnel lenses are typically used in situations in which refractive focusing is difficult to perform or planar fabrication is advantageous. However, the geometrical complexity and low efficiency of Fresnel lenses impair their applications in practice[3]. Furthermore, the focusing capability of these bulk lenses deteriorates when their dimensions are reduced toward wavelength-sized scale.

The recent emergence of a family of planar metamaterials, named metasurfaces, has attracted significant attention[4–21]. Metasurfaces break the propagation dependence by controlling the wavefront with subwavelength-spaced structures (i.e. unit elements), and they have shown considerable potential in achieving similar or improved functionalities in wave manipulation while significantly reducing the dimensions. The outstanding characteristics of the metasurfaces (i.e. wavefront control with subwavelength resolution, thin and light form factor, and compatibility with standard deposition techniques) make them very attractive for developing modern devices, particularly in the acoustic regime, as the wavelength is much larger than that in optics. Acoustic metasurfaces use arrays of unit elements with spatially varying geometric parameters (e.g. shape, size, and depth) to form a spatially varying acoustic response, molding the wavefronts into desired shapes. Acoustic responses can be controlled by engineering the wave interactions with the unit elements, which can take various forms, such as masses deposited on films and apertures opened in thin plates. Based on this concept, acoustic metasurfaces have exhibited diverse functionalities, including focusing[8–10], extraordinary transmission and reflection[11,12], negative refraction[13], one-way propagation[14–17], perfect absorption[18,19], and cloaking[20,21]. In the development of wave manipulation with subwavelength thickness, coiled space elements, including zig-zag[8–11,22], labyrinthine[13,23], and helical structures[24] have been extensively exploited. In conjunction with 3D printing, these materials have shown promising potential for manipulating airborne sound in architecture acoustics and related fields[21,25]. However, this scheme has not allowed the creation of ultra-compact devices because the elements are either equipped with a considerable thickness to yield an extreme refractive index, or they are assembled in a diffraction-limited manner, to simplify the design process by treating individual unit elements without the consideration of the couplings between them. On the other hand, given their complex geometries and narrow channels, they inevitably cause fabrication difficulties, particularly when their dimensions become smaller[20,26]. Furthermore, they face challenges in direct manipulation of acoustic fields, especially the near-fields, by controlling the effective path length. Therefore, creating acoustic metasurfaces without cumbersome structures would be of both fundamental and practical significance, but the complexity of the current structures remains a critical barrier.

In this paper, we propose another avenue in the design of the acoustic metasurface lens, to extend the wave manipulations to both far- and near-fields and markedly reduce the design

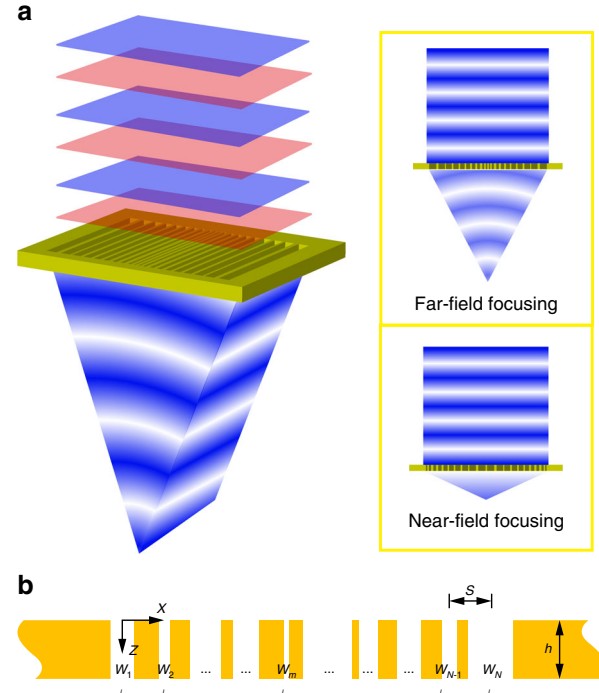

**Fig. 1** Metasurface-based acoustic lens. **a** Convergence of acoustic energy by the metasurface lens after a normally incident plane wave transmits through it. Through proper design of the individual unit elements and their arrangement, different focusing schemes can be achieved, as presented in the insets. **b** Schematic representation of the acoustic metasurface lens. It consists of $N$ slits that are periodically perforated in a thin plate with a spacing of $s$ and a depth of $h$. The width and phase shift of the $m$th slit are denoted by $w_m$ and $\phi_m$, respectively

cost and implementation complexity, using a simple structure (schematically depicted in Fig. 1), which consists of a periodic array of slits in a thin plate, with slit spacing and plate thickness both very much smaller than the wavelength. In general, the acoustic response of each slit is individually tuned by adjusting its size[27,28], localized resonance[29,30], or filling material[9,31]. The wave couplings between the slits can be neglected if they are deep enough and therefore the slits are considered to be independent. However, this is not the case when the thickness of the plate, and thus depth of the slits, is much smaller than the wavelength. For this reason, the simple design rule based on the properties of individual slits would result in a significant discrepancy in focusing. Such discrepancy becomes more severe when spacing of the slits is also much smaller than the wavelength. Several studies have investigated the causes of the focal shift effect, but they are mainly focusing on the influences of the structural parameters, such as slit number, focal length, and lens width[32,33]. Indeed, these parameters will affect the focusing behavior, but the underlying physics on how they change the wave dynamics and modify the field distribution remain to be clarified. Meanwhile, the simple design rule faces another difficult challenge that the tuning capability for a single slit becomes very poor if both its width and thickness are deeply subwavelength. It is also worth noting that the periodic subwavelength patterned structures have been widely exploited in optics and fluid dynamics, and effective medium theory[34] is generally employed to study their macroscopic properties as they are often believed to be governed by near-field interactions. However, this theory also cannot be applied to our design. The reason is that, as will be shown hereafter, coupling effects inherent to multiple-scattering

processes are inevitable and significantly affect the acoustic diffractive pattern of each slit.

Thus such an acoustic metasurface lens may seem unfeasible to be realized because of the lack of a design strategy. To this end, we develop a microscopic semi-analytical approach for the lens design by fully considering the wave dynamics on the metasurface. In the microscopic description, we show that the diffractive acoustics of a slit can be controlled not only by its own acoustic response but also by the other slits through strong couplings. For proof-of-concept, we examine different schemes of sound focusing by optimizing the metasurfaces with our microscopic model. The capability to pack simple structures at spacing well below the wavelength suggests a feasible approach for creating ultra-compact acoustic components with ease-of-manufacture and versatile potentials.

## Results

**Microscopic coupled-wave model.** As shown in Fig. 1b, the metasurface lens was composed of a thin plate perforated with $N$ straight slits with center-to-center spacing $s$ and plate thickness $h$. A plane wave with a wavelength of $\lambda$ was impinged on the metasurface. The width of the $m$th slit was denoted by $w_m$ ($m = 1 \ldots N$), and the phase shift of the plane wave passing through the $m$th slit was denoted by $\phi_m$. In our scheme, both the spacing and the thickness were on the subwavelength scale. In this scenario, the slits cannot yield an extreme refractive index and thus failed to localize the acoustic field to the interior[35]. Therefore, multiple-scattering cannot be neglected and can be observed whenever a set of subwavelength apertures was arrayed. Such scattering processes may modify the acoustic diffractive pattern of the slits and in turn affect the distribution of acoustic fields.

To confirm the existence and evaluate the importance of coupling effects, we started with the simplest system, that is, a slit doublet (Fig. 2a). While the double-slit experiment is well-known in optics and water waves, it has not been attempted yet when the slits are separated with a deep-subwavelength distance, with which the coupling effect becomes prominent. For convenience, we studied the coupling effect using a numerical simulation model (see Methods).

Hereafter, we referenced the left slit and set its parameters as $h = 0.05\lambda$ and $w_1 = 0.03\lambda$. At a normal plane wave incidence, the total field at the exit of the reference slit was the sum of the direct transmission and the coupled field from the neighboring slit. The coupling effect can be obtained from the total field by subtracting that associated to a single slit. We define this effect as $\Delta P = P' - P_0$ and $\Delta \phi = \phi' - \phi_0$, where $P'$ and $\phi'$ are the amplitude and phase of the pressure field extracted at the exit of the reference slit in the doublet configuration and $P_0$ and $\phi_0$ are those from the single reference slit. For simplicity, the pressure of the incident plane wave was normalized to 1 Pa. The direct transmission of the reference slit was numerically computed to be $0.611 - 0.125i$ Pa, that is $P_0 = 0.624$ Pa and $\phi_0 = -0.2$ radians. Figure 2b, c shows the results of $\Delta P$ and $\Delta \phi$ by varying the width of neighboring slit and the separation distance between them. For clarity, $\Delta P$ and $\Delta \phi$ as a function of slit width ($d = 0.2\lambda$) and separation ($w_2 = 0.03\lambda$) are plotted in Fig. 2d, e, respectively. As predicted, the coupled pressure amplitude and phase varied with the width of adjacent slit and the distance between them, which indicated a direct signature of the coupling effect. This change resulted from the wave funneling and scattering between these two slits, as shown in Fig. 2a. The incident wave scattered by a slit can be regarded as a new point source that excited the cylindrical wave (CW), which was the superposition of the evanescent and propagation fields. The CW on the surface was

partially funneled into the other slit as well as scattered by it. The scattered and funneled waves then excited the CWs again at the slit ports. The processes of scattering and funneling were repeated until the CWs on the surfaces were diminished. Therefore, quantitative information on the amplitude and phase of the coupled field can be obtained by considering the whole wave dynamics.

To explore this issue, we developed a microscopic coupled-wave model that considered the total field as the sum of the scattered waves by every slit, which involved the transmission of the initial incidence as well as the funneling and scattering of the excited CWs. This strategy required knowledge of the acoustic response of a subwavelength slit. Thus, we began with the fundamental process that described the wave interaction with a single slit and then analytically calculated the transmission and reflection coefficients by using the coupled-mode theory[36]. Under a uniform plane wave incidence, the acoustic fields above and below the slit can be expressed in terms of plane wave expansions. In the absence of cutoff frequency, acoustic waves can freely propagate within the subwavelength slit. By contrast, only the fundamental mode is supported under the subwavelength condition (i.e., $w \ll \lambda$ with $w$ being the slit width). Therefore, the acoustic field inside the slit can be well described as a superposition of two counter-propagating waves[37]. According to the boundary continuities, the pressure field and normal velocity of the fundamental waveguide mode should match the plane wave expansions outside of the slit at both interfaces. Consequently, the transmission and reflection coefficients for the normal (Fig. 2f) and grazing (Fig. 2g) incidences can be obtained as $T_n = \frac{4Qe^{ik_0h}}{(Q+1)^2 - (Q-1)^2 e^{2ik_0h}}$, $R_n = \frac{Q-1}{Q+1}\left(T_n e^{ik_0h} - 1\right)$, $T_g = \frac{2Qe^{ik_0h}\mathrm{sinc}(k_0w/2)}{(Q+1)^2 - (Q-1)^2 e^{2ik_0h}}$, and $R_g = \frac{T_g e^{ik_0h}(Q-1) + \mathrm{sinc}(k_0w/2)}{Q+1} - 1$, where the subscript n(g) indicates the normal (grazing) incidence; $k_0 = 2\pi/\lambda$ is the acoustic wavenumber; sinc is the unnormalized cardinal sine function; and $Q = \frac{k_0}{\pi w} \int_{-\infty}^{\infty} \frac{1 - \cos(w\alpha)}{\alpha^2 \sqrt{k_0^2 - \alpha^2}} \, d\alpha$ represents the coupling between the fundamental waveguide mode and all diffractive waves[38] (Supplementary Note 1). These analytical expressions were quantitatively verified on different slit parameters with numerical simulations that considered the visco-thermal loss (Supplementary Fig. 2). The good agreements between the calculations and the simulations confirmed that they were adequate for describing the waves scattered by a subwavelength slit.

Based on this result, we then considered the CWs. For a scattered CW spreading out from a subwavelength slit, its distribution can be described as $A(r) = \beta e^{i\phi} H_0^{(2)}(k_0 r)$[39], where $\beta$ and $\phi$ are the scattering coefficient and abrupt phase jump of the CW; $r$ is the distance from the slit center and $H_0^{(2)}$ is the zeroth-order Hankel function of the second kind. Accordingly, the pressure field of the excited CW can be written as

$$P_{\mathrm{sca}}(r) = P_{\mathrm{inc}} S_{\mathrm{sca}} A(r) = P_{\mathrm{inc}} S_{\mathrm{sca}} \beta e^{i\phi} H_0^{(2)}(k_0 r), \tag{1}$$

where $P_{\mathrm{inc}}$ is the pressure of the incident plane wave; $S_{\mathrm{sca}}$ denotes the transmission or reflection coefficient ($T_n$, $R_n$, $T_g$, or $R_g$).

To assess the relationship between the CWs and the slit parameters, we adopted a two-step semi-analytical procedure to calculate $\beta$ and $\phi$. First, we performed numerical simulation on a single slit illuminated by a plane wave and extracted the pressure field along the surface. The pressure field can also be acquired from experimental measurements; however, this approach is time-consuming and may introduce measuring errors. A numerical-based method is more viable. In the second step, using Eq. (1), we fitted the pressure field distribution over an

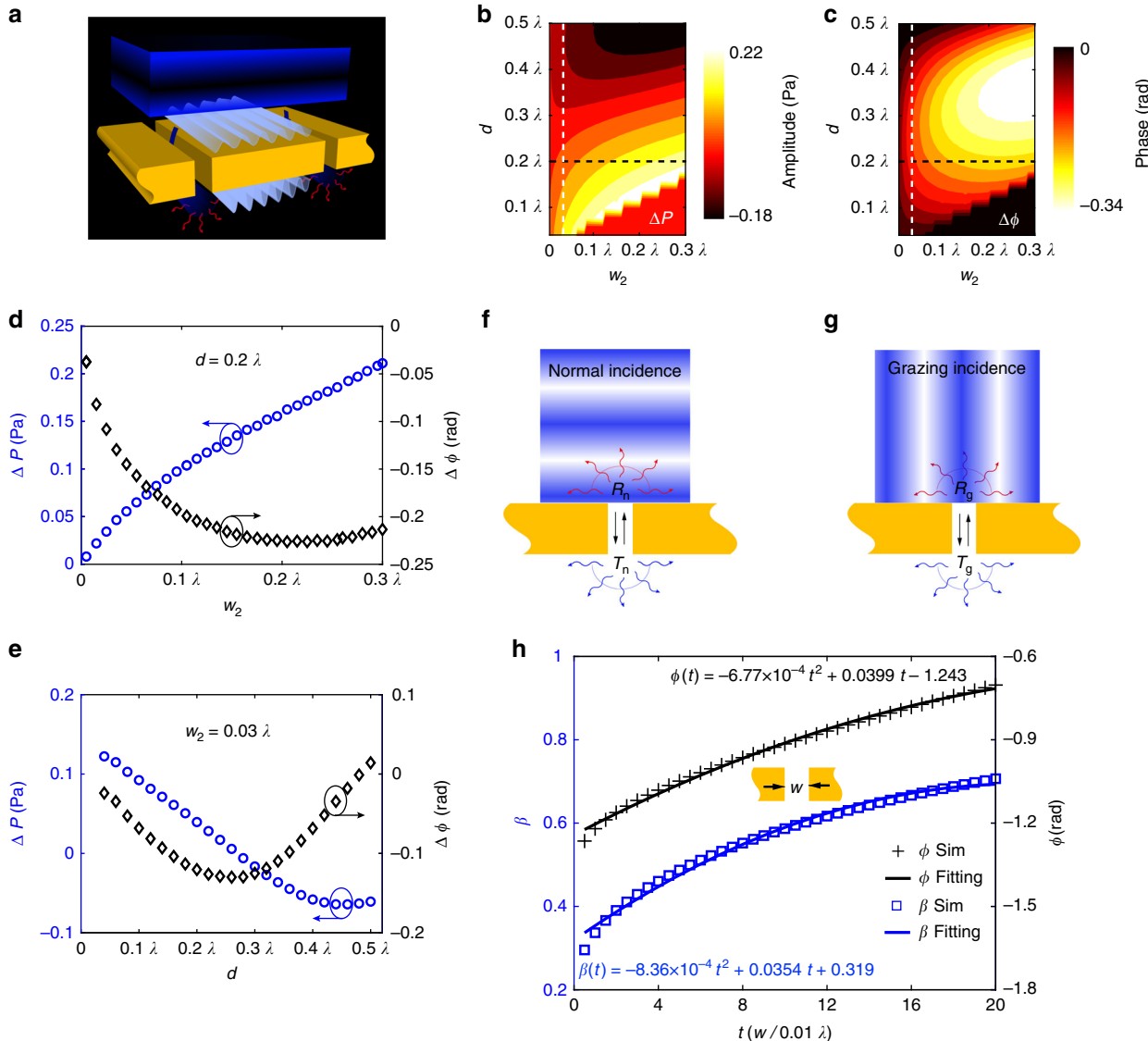

**Fig. 2** Microscopic coupled-wave model. **a** The wave dynamics on the slit-doublet configuration. The incident wave scattered by a slit can be regarded as a new point source which then generates a cylindrical wave (CW). The CWs on the surface are partially funneled into the other slit, as well as scattered by it. The scattered and funneled waves again excite the CWs. The processes of scattering and funneling are repeated until the CWs are diminished. For clarity, only one iteration process is depicted. **b**, **c** The changes of amplitude ($\Delta P$) and phase ($\Delta\phi$) at the exit of the reference slit affected by the width $w$ of adjacent slit and the distance $d$ between them. **d** The profiles of $\Delta P$ and $\Delta\phi$ as a function of slit width along the black dashes ($d = 0.2\lambda$) in Fig. 2b, c. **e** The profiles of $\Delta P$ and $\Delta\phi$ as a function of separation distance along the white dashes ($w_2 = 0.03\lambda$) in Fig. 2b, c. **f**, **g** The transmission and reflection coefficients of a subwavelength slit for normal and grazing incidences. **h** The dependence of the characteristic parameters ($\beta$, $\phi$) of the CW on the slit width. They are fitted as a polynomial function using the least-mean-square method. The slit width is scaled in the fitting, i.e., $t = w/0.01\lambda$. The source data underlying Fig. 2h are provided as a Source Data file

interval ($w/2 \leq r \leq 10\lambda$) by optimizing the unknown parameters. Repeated calculations were conducted for various slit parameters (Supplementary Note 2). These calculations revealed that the characteristic parameters of the CW were associated with slit width only and were independent of the thickness (i.e. $A(w, r)$). To link the characteristic parameters of the CW to the slit width, $\beta$ and $\phi$ are fitted as a polynomial function with the least-mean-square method and shown in Fig. 2h. Note that the slit width is scaled in the fitting, i.e., $t = w/0.01\lambda$. It is also worth noting that the fit does not coincide well when the slit width approaches zero, and a small deviation can be seen at $t = 0.5$. Nevertheless, we think this will not be a big problem as a lower bound is generally set on the slit width in the metasurface design to facilitate the fabrication (see Methods).

After the acoustic response of a single subwavelength slit was determined, we proceeded with the quantification of the coupled fields. Although coupled fields can be evaluated using simulation-based numerical studies, these approaches are tedious and unable to elucidate the underlying physics, such as which components of the coupled fields would affect the diffractive acoustics and what their weights would be in the total contribution. To gain further insight, we formulated the microscopic coupled-wave model by fully considering the wave dynamics on the deep-subwavelength structures (see Methods). This model worked not only as a straightforward routine that reproduced the field distribution after an incident wave passed through a designated metasurface (Supplementary Fig. 3), but also as a design recipe that optimized the acoustic metasurface for desired wave manipulation.

**Metasurface lens for far-field sound focusing.** As the most basic and important functional element, the acoustic lens plays a crucial role in the far-field focusing of acoustic energy in various applications. To ensure all waves arrive in phase for constructive interference at the focal spot, the phase profile of the diffractive waves on the exit surface, as predicted by Huygens' Principle, should follow

$$\varphi(x) = \frac{2\pi}{\lambda}\left(f - \sqrt{(x-x_0)^2 + f^2}\right), \qquad (2)$$

where $f$ is the focal length and $x_0$ is the center of the lens. To demonstrate the proof-of-concept, we considered a metasurface lens operating under transmission mode. Without loss of generality, the dimensional quantities were scaled to the wavelength in the following context. The focal length and numerical aperture (NA = $D/2f$ with $D$ being the overall aperture) of the metasurface lens were set to $3.0\lambda$ and 0.5, respectively. According to the settings, the acoustic metasurface was arranged with 19 slits (i.e., $N = 19$), which were equidistantly separated with a spacing of $0.17\lambda$. The thickness can be arbitrarily opted in the deep-subwavelength range, as the single-pass phase delay is negligible. Here, a thickness of $0.05\lambda$ was assigned. Given the assigned structural parameters and the established microscopic model, the slit widths in the acoustic metasurface were numerically optimized with nonlinear least-squares fitting (see Methods), which minimizes the differences between the actual phase across the exit surface and that from Eq. (2). The optimized slit widths are listed in Supplementary Table 1.

To validate the result, numerical simulation was performed on the optimized metasurface lens. Figure 3b shows the simulated intensity distribution of the pressure field after a plane wave was transmitted through the metasurface lens, in which a focal spot was clearly observed at the focal depth (white dashed line). For completeness, the calculated intensity distribution using the microscopic model is illustrated in Fig. 3a, which was in good agreement with the simulated image. To quantitatively characterize the focusing performance, the profiles of the pressure amplitude along the transverse and axial directions through the focal spot are shown in Fig. 3d, e. As shown, the profiles coincided well and nearly overlapped with each other. The full-width at half-maximum (FWHM) and focal length were measured to be $0.98\lambda$ and $2.9\lambda$, respectively. Both results were consistent with their prediction (the focal spot size BW is calculated to be $1.02\lambda$ with BW = $0.51\lambda$/NA). Meanwhile, the incident and transmitted energy were numerically computed by integrating the intensity over the transmission domain ($-2\lambda \leq x \leq 2\lambda$, $0 \leq z \leq 8\lambda$), which were 37.8 and $17.3\,\mu$W, respectively. Thus, a transmittance of 45.8% can be obtained for the designed metasurface lens. Accordingly, these simulation results provided a direct visual verification of the design rule.

For further confirmation, experimental measurements were conducted on the metasurface sample fabricated with the optimized parameters (see Methods). The measured intensity distribution and pressure amplitude profiles across the focus are plotted in Fig. 3. As shown in the figures, good agreements can be observed among the calculation, the simulation, and the experiment, further verifying the effectiveness of the metasurface design. In particular, the focal lengths were well matched, thereby effectively demonstrating the accuracy of wave manipulation. Note that the distribution of the measured pressure field was not as uniform as the simulation and calculation. This was mainly because of the coarse sampling in the $z$ direction. Also, it was affected by the diffractions of the incident wave from the outer edges of the metasurface lens because of its finite width. The presence of such diffractions was testified by the slight amplitude

rise of the sidelobes in experiment in Fig. 3d. Meanwhile, the measured FWHM is a bit larger than the calculated and simulated FWHMs, and this discrepancy arose from the convolution of the focusing pattern with the collection volume of the measuring microphone. It is worth noting that the calculated and simulated results were obtained using the optimized parameters rather than the actual ones measured from the sample. This agreement demonstrated the robustness in the metasurface lens design and the tolerance in fabrication.

In addition, we studied the focusing behaviors of the optimized metasurface lens with respect to thickness while keeping the other parameters unchanged. The simulated FWHM and focal length with different thicknesses are shown in Fig. 3f. Although the transmission changed with the thickness, as shown in the insets, the FWHM and focal length were nearly the same, whereas the thickness varied in the range of $[0.01\lambda, 0.1\lambda]$. This can be expected as only phase is taken into account in the far-field sound focusing. In other words, we can reduce the metasurface thickness down to an order of magnitude smaller than the state-of-the-art ultrathin lens by coiling up space[11]. Also, for a given focal length, the focal spot size depends on the overall aperture of the metasurface lens: the larger the aperture, the higher the NA, and the tighter the focus. According to the Rayleigh criterion, the far-field lens would give a diffraction limit of $0.61\lambda$, whereas the FWHM of the focal spot in simulations was found to be approximately $1.0\lambda$. It is worth noting that although the focal spot of the far-field lens can be further narrowed by increasing the overall aperture, it will be restricted eventually by the diffraction limit due to the loss of evanescent waves. To break this limit and achieve sub-diffraction focusing, the evanescent waves, which are bound to the near field, should be incorporated.

**Near-field lens for patterned sub-diffraction focusing.** At present, sub-diffraction focusing has received considerable interest because of its close relation to super-resolution imaging. As mentioned, the near-field evanescent waves should be incorporated and delivered to the target focus to create a sub-diffraction focus. Our scheme provided a conceptual advantage for sub-diffraction focusing, as the unit elements were deep-subwavelength sized and arranged. Moreover, the microscopic model can accurately reproduce the near fields because no assumptions such as far-field approximation were made in the derivation. We demonstrated a metasurface-based near-field lens for sub-diffraction focusing through the direct manipulation of evanescent waves.

To produce a desired pattern at the focal plane, that is, $P(x, z = f)$, the first step is to find the field distribution on the exit surface of the near-field lens, that is, the aperture field, through back-propagation[40]

$$P(x, z=0) = \frac{1}{2\pi}\iint P(x', f)e^{i[k_x(x-x')-k_z f]}\,dx'\,dk_x \qquad (3)$$

where $k_x$ and $k_z = \sqrt{k_0^2 - k_x^2}$ are the wave vectors along $x$ and $z$ directions, respectively. As a particular design, the focal length and pattern were defined as $0.1\lambda$ and $P(x, z=f) = M_0 e^{-\left(\frac{x-x_0}{2\sqrt{\ln 2}\cdot\text{FWHM}}\right)^2}$, where $M_0$ is an amplification factor and $x_0$ represents the lens center. Figure 4d shows the profiles of pressure amplitude at the aperture plane ($z = 0$) and at the focal plane for the case of $M_0 = 2$ and FWHM = $\lambda/6$. Then, we set the thickness, spacing, and slit number to $0.38\lambda$, $0.1\lambda$, and 11, respectively. Note that the thickness of the near-field metasurface lens was set beyond the deep-subwavelength scale. This is because, for the sub-diffraction focusing with a loose spot at a considerable depth, the contributions mainly came from the evanescent waves of low spatial frequencies, and a relative large

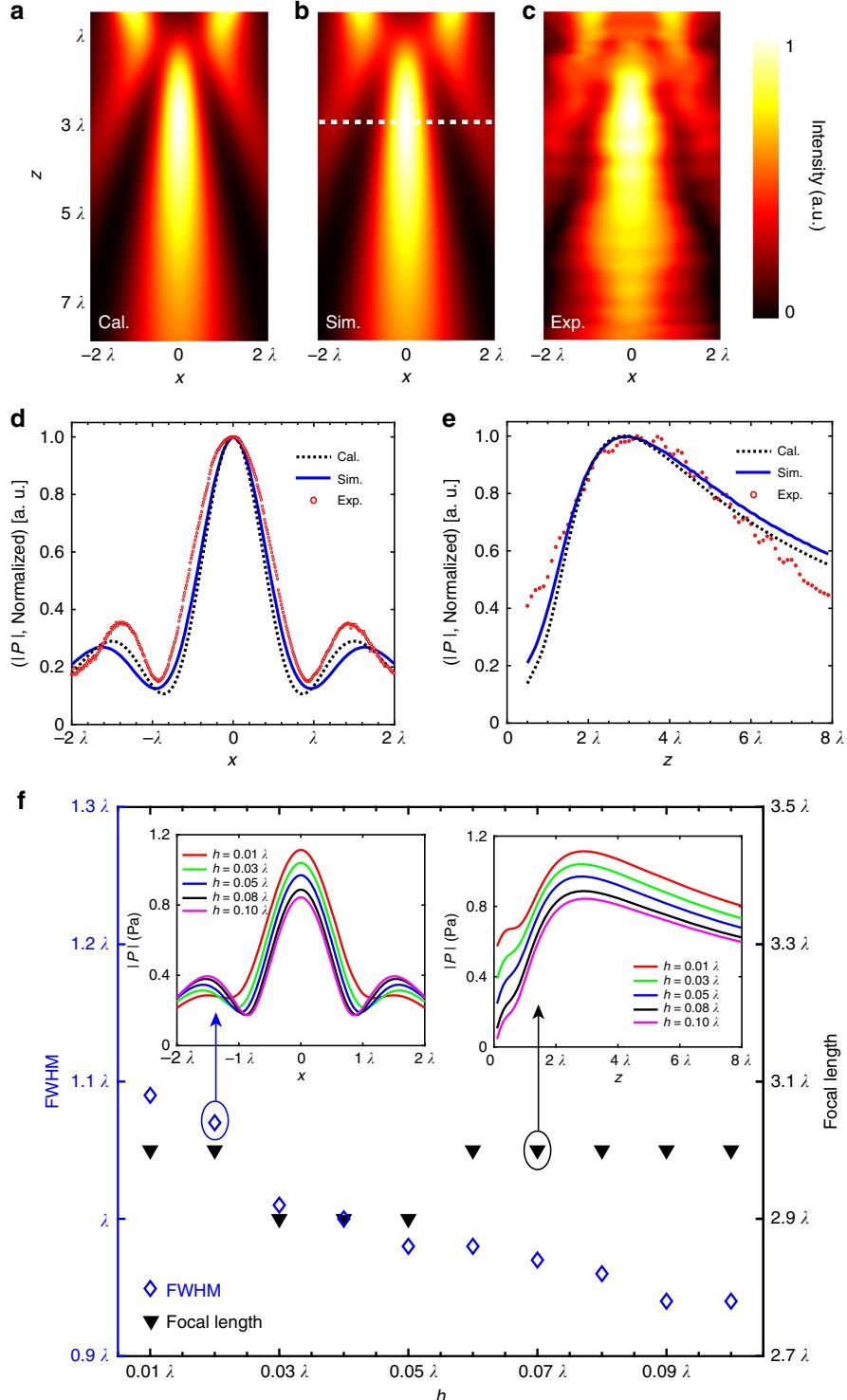

**Fig. 3** Far-field acoustic focusing. **a–c** Theoretical calculation, numerical simulation, and experimental measurement of the pressure field, showing the normalized intensity maps after a normally incident plane wave transmits through the optimized metasurface lens. The white dashed line denotes the designated focal length. **d**, **e** The distributions of normalized pressure amplitude in cross sections of the focal spot along $x$ direction and $z$ direction, respectively. **f** Simulated full-width at half-maximum (FWHM) and focal length with different thicknesses of the acoustic metasurface varying in the range of $[0.01\lambda, 0.1\lambda]$. The profiles of pressure amplitude through the focal spot are shown in the insets

thickness was employed to efficiently couple these evanescent waves from the input surface to the output side. In contrast to the optimization on the phase profile for far-field focusing, the near-field metasurface lens was optimized to yield the aperture field, which in turn produced the focusing pattern at the focal plane.

Figure 4a–c show the calculated, simulated, and measured intensity distributions of the pressure field with the optimized near-field metasurface lens (Supplementary Table 2) impinged by a normal plane wave. The field mapping images show good agreement, and sub-diffraction focal spots are clearly

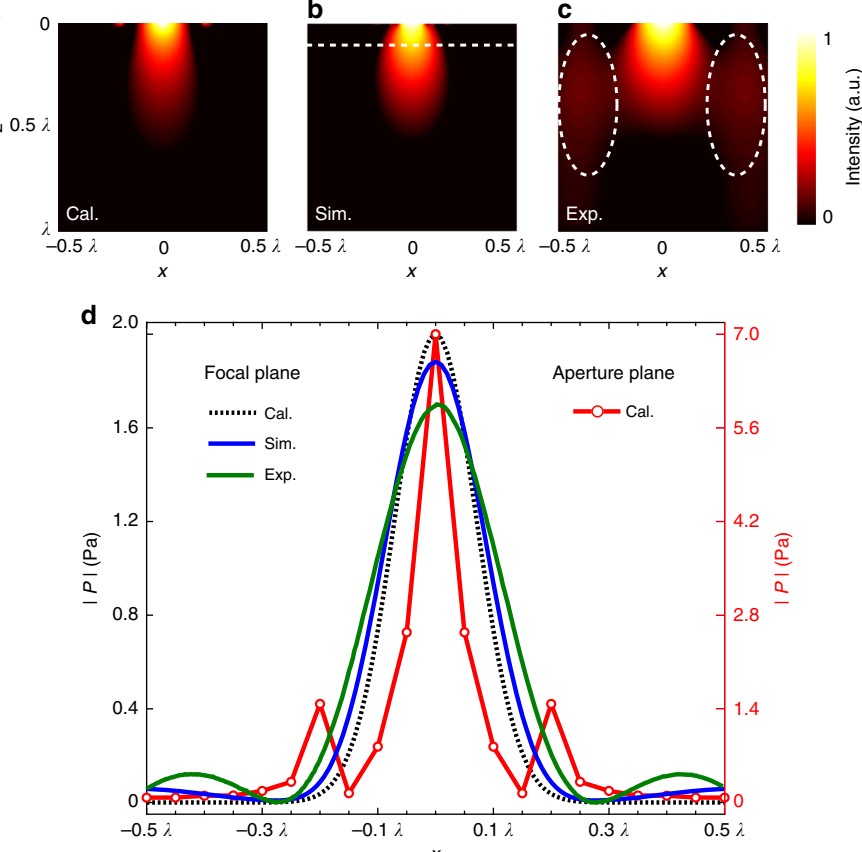

**Fig. 4** Near-field sub-diffraction focusing. **a–c** Normalized intensity distributions of the calculated, simulated, and measured pressure field, showing the sub-diffraction focusing under the normal plane wave incidence. The white dashed line denotes the focal plane. The differences between the theory and the experiment are circled out. **d** The profiles of the pressure field amplitude at the aperture plane ($z = 0$) and at the focal plane ($z = 0.1\lambda$)

demonstrated, verifying that the designed near-field lens can effectively manipulate the evanescent waves as expected. However, the measured sub-diffraction focusing was slightly broader than those from calculation and simulation. This cause was mainly attributed to the diffractions from the edges due to the finite size of the metasurface sample. For the same reason, some unwanted sidelobes can be seen distributed around the vertical boundaries (white dashed ellipses). Meanwhile, as mentioned above, it is expected that the microphone widened the focal spot and the fabrication tolerances in 3D printing also affected the measured field distribution. To quantify the focal pattern, the pressure amplitude profiles along the transverse direction at the focal plane (white dashed line) are also plotted in Fig. 4d. As shown, the pressure amplifications in the simulation and experiment were 1.9 and 1.7, respectively. These results are consistent with the assigned one ($M_0 = 2$), although a bit smaller due to the wave dissipation effect. Similarly, the simulated and measured FWHMs were obtained to be $0.19\lambda$ and $0.23\lambda$, which are close to the designation ($\lambda/6$) but slightly larger. However, this deviation can be anticipated as the aperture field was only matched at ($N =$) 11 discrete points (slit centers). In addition, as the slits were simplified as point sources in the optimization, the actual focusing behavior would be somewhat deviated from the calculation, considering that a wider slit typically makes more contribution for the focusing. Particularly, the measured FWHM was further enlarged because of the measuring microphone ($0.02\lambda$ in diameter). Nevertheless, we consider the result to be satisfactory and the deviations can be accounted through numerical simulations.

Thus the above results readily prove the effectiveness of the design rule in sub-diffraction focusing. In particular, the focusing patterns can be tailored to produce other types and symmetries under the same procedure. Furthermore, the direct control of the acoustic near field presented here may be helpful in exploiting other potentials in the subwavelength regime.

## Discussion

In conclusion, we introduced a comprehensive design rule for the metasurface lenses to achieve acoustic focusing by suitably tuning the acoustic diffractive pattern of slits that are perforated in a thin plate with deep-subwavelength spacing. The design rule was developed based on a microscopic coupled-wave model, which considered the whole wave dynamics on the deep-subwavelength structures, which showed that the total field was the sum of the scattered waves by every slit, suggesting that the direct transmission and the secondary scatterings of coupled fields from other slits were involved. To show the effectiveness of the proposed design rule, we calculated theoretically, proved numerically, and demonstrated experimentally two examples of metasurface lenses under transmission mode; these yielded diffraction-limited focusing in the far field and mimicked a near-field plate for patterned sub-diffraction focusing. Compared with the previous design rules for the slit structure, such as using single-pass phase retardation[41] or simulation-based numerical optimization[42], the semi-analytical approach presented here worked in a simple, straightforward and accurate manner, and achieved a much better performance on the prediction of the focusing behavior.

Remarkably, we can directly manipulate the acoustic waves in both far and near fields.

Compared with the popular space-coiling acoustic metasurfaces, the proposed one offers the advantages of ultra-compact form, simple design and easy fabrication. In particular, the thickness of far-field metasurface lens can be an order of magnitude smaller than the state-of-the-art ultrathin counterpart by coiling up space. Nonetheless, this significant reduction in thickness somewhat sacrificed the overall phase tuning capability, and the phase match became progressively difficult for increasing phase gradients. The improvement on the phase tuning range beyond the thickness dependence will be studied in future works. Moreover, it is necessary to point out that the overall aperture of the far-field metasurface lens is limited, that is $(N-1)\,s \leq 2f$. Also, a limited overall aperture is sufficient for near-field focusing as the evanescent waves are exponentially decaying and the contributions from outer slits decrease rapidly when they move away from the lens center.

Considering that the acoustic response of the deep-subwavelength structures and the whole wave dynamics between them were accurately characterized, the proposed microscopic model can also be used for other forms of wave manipulation, such as focusing of surface grazing waves, beam steering, and asymmetric transmission. Meanwhile, it could be adapted easily to the study of reflective metasurfaces with groove structures for the full control of reflected waves. Other studies on the metasurface lens mainly involve optimization for better performance (such as aperiodic distribution of slits[43] and stacking of multiple metasurfaces[10,44]), frequency response, and potential use of amplitude and phase modulation[45,46]. These further studies are beyond the scope of this work.

Scaling to the sizes of interest, we predict that the metasurface lens can be realized in the ultrasonic regime, and that their outstanding features make them very attractive for integration with microelectromechanical systems technology[47]. Moreover, this lens shows distinct advantages in the very long wavelength regime with the ultra-compact arrangements. Given that similar structures are also found in nanophotonics[41,48], our results suggest potential for the analogous design of transmissive flat optical lenses, which are crucial in compact size and on-chip optoelectronic integration.

## Methods

**Numerical simulations**. Numerical simulations were performed using the commercial finite element (FEM) software COMSOL Multiphysics 5.1. All simulations were conducted in 2D. Modules of pressure acoustics and solid mechanics were utilized to study the wave-structure interaction, and the acoustic-structure boundary interface accounted for the coupling between different physics. The geometry was modeled with the assigned and optimized structural parameters. Perfect matched layers were added to the outer boundaries of the simulation domain to mimic an anechoic environment. A background pressure field was used for plane wave generation. Subwavelength slits were attributed to Narrow Region Acoustics. The mesh element size was set to be smaller than $\lambda/12$, and a refined mesh was applied inside the slits to account for the fine features at the solid–air interfaces. The simulated acoustic fields were exported for further analysis and processing.

**Experimental setup and acoustic field measurements**. Field mapping measurement was obtained using a calibrated multi-field B&K microphone (type-4961) and a lab-made 2D scanning stage in an anechoic room. The wavelengths (operating frequencies) for the far-field and near-field lenses were set to 100 mm (3.43 kHz) and 300 mm (1.15 kHz), respectively. The far-field lens was fabricated by cutting 30 cm-long slits on an aluminum plate (600 mm × 500 mm × 5 mm), and the near-field plate was 3D printed using ABS-M30 (Stratasys, USA). A planar speaker (60 × 60 sound shower; Panphonics) was used for the plane wave generation (Supplementary Fig. 4 and Supplementary Table 3). A tone-burst (15-cycle) signal was provided by a USB sound card (sound blaster X-Fi; Creative Technology, Singapore). The speaker was aligned in parallel to the sample and placed at a distance of 1.2 m. During the experimental measurements, the microphone was raster scanned to record the pressure field in the transmission region. The scanning

step is 1 mm in $x$ direction for both measurements. For the $z$ direction, the step is 10 mm for the far-field lens and 3 mm for the near-field plate. The output signal of the microphone was preamplified with a B&K signal conditioner (model 1704-A-001) and digitized by the sound card, and stored in a computer for further processing. The loop-back signal from the speaker was also recorded. The schematic of the experimental setup is depicted in Supplementary Fig. 5.

**Formulation of microscopic coupled-wave model**. For a normally incident plane wave, the pressure fields of the initial reflection and transmission by each slit can be expressed as $P_{sca}^{a}(0,m) = P_{inc}R_{n}(w_{m})$ and $P_{sca}^{b}(0,m) = P_{inc}T_{n}(w_{m})$, where superscript a (or b) denotes the top (or bottom) port of the slit; $R_{n}(w_{m})$ and $T_{n}(w_{m})$ are the reflection and transmission coefficients of the $m$th slit with a width of $w_{m}$ ($m = 1, 2 … N$). The scattered waves at the slit ports excited the CWs, and the component along the surfaces were re-scattered and funneled by the other slits. This process was iterated until the CWs scattered on the surfaces were diminished. At each iteration, the summation of the coupled pressure field at the $m$-th slit from all other slits can be expressed as

$$P_{cpl}^{a}(l,m) = \sum_{\substack{j=1 \\ j \neq m}}^{N} P_{sca}^{a}(l,j) A\left(w_{j}, |m-j|s\right) \tag{4}$$

$$P_{cpl}^{b}(l,m) = \sum_{\substack{j=1 \\ j \neq m}}^{N} P_{sca}^{b}(l,j) A\left(w_{j}, |m-j|s\right) \tag{5}$$

where $l$ represents the iteration index. In view of the funneling and scattering process, the diffractive pressure field at the slit top can be written as

$$P_{sca}^{a}(l+1,m) = P_{cpl}^{a}(l,m)R_{g}(w_{m}) + P_{cpl}^{b}(l,m)T_{g}(w_{m}) \tag{6}$$

and that at the slit bottom as

$$P_{sca}^{b}(l+1,m) = P_{cpl}^{a}(l,m)T_{g}(w_{m}) + P_{cpl}^{b}(l,m)R_{g}(w_{m}) \tag{7}$$

Finally, the diffractive pressure field at the bottom port of each slit can be given as

$$P^{b}(m) = \sum_{l=0}^{Y} P_{sca}^{b}(l,m), \tag{8}$$

where $Y$ is the iteration number. According to Eq. (8), the pressure field and phase profile on the lower surface of the metasurface can be accurately predicted. We explicitly verified the validity of the formulated microscopic model by comparing the theoretical calculation and the numerical simulation on a user-defined metasurface. The calculated and simulated results were in quantitative agreement (Supplementary Fig. 3).

**Numerical optimizations**. Numerical optimizations were performed by using the nonlinear least-squares solver in Matlab 2013b. The error function for nonlinear fitting was user-defined by implementing Eqs. (4)–(8). The iteration number was an important concern for optimization efficiency. In this paper, the iteration number was 135 (far-field focusing) and 210 (sub-diffraction focusing). The minimum iteration number typically depended on the slit number $N$ and spacing $s$, and can be evaluated by performing a straightforward calculation with the initial guess of slit widths. To facilitate the fabrication of the metasurface lenses, a lower bound was set on the slit width in the optimization of both far-field lens ($t = 2.0$) and near-field plate ($t = 0.3$).

## Data availability

The data supporting the results of this study are available within the published article and Supplementary Information. The source data underlying Fig. 2h are provided as a Source Data file. Further information is available from the corresponding author upon reasonable request.

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

## Acknowledgements

We would like to thank Prof. Michael Lowe from Imperial College London for the critical reading of the paper and useful suggestions. We would like to thank Dr. Bin Liang from Nanjing University and Dr. Likun Zhang from The University of Mississippi for fruitful discussions.

## Author contributions

J.C. and Z.F. conceived the idea. J.C. and J.X. performed the theoretical calculations and the numerical simulations. J.C., D.L. and A.S. fabricated the sample, performed the experimental measurements and data analysis. Z.F. supervised the whole project. J.C and Z.F. wrote the manuscript, and all the authors reviewed the manuscript.

## Additional information

**Competing interests:** The authors declare no competing interests.

