## [Peer Review File · Nature Communications]

Reviewers' Comments:

Reviewer #1:

Remarks to the Author:

I think it's an interesting paper, and suitable for publication in Nature Communications. Technically it is well argued, the methods are clear, and there is enough information supplied that one could reproduce results. The design method is definitely semi-analytical in that you need a fairly sophisticated code to extract the complex amplitude of the cylindrical waves (beta and phi). But with this info, it allows a design approach that is otherwise harder to implement by brute force.

My only concern is whether the significance of the results are a bit over-stated (from a practical standpoint). Specifically, the focus that is observed in Figure 3b is only a doubling in amplitude from the main lobe to the first side-lobe. So that's not a strong focus. Given that the slits are so deep-sub-wavelength in width and thickness, it is a bit surprising that the wave fields can conspire to produce even that much of a focus. In other words, features so small generally do not create large scattering unless resonance is in play. But the reasoning that significant phase modification is coming from "circulation" (repeated funneling and multiple scattering), explains the effect well. But it is unfortunate that a computational case is not presented with a wider overall aperture to test the limits of the approach.

Another general comment has to do with the applicability of the term "Fresnel lens". The "Fresnel" aspect is a bit of a misuse of the term in a sense that a Fresnel zone concept is not at play. But there are slits in a plate making an aperture so it is a bit related to Fresnel lenses. Also, the authors mention low focusing efficiency of Fresnel lenses, but I don't believe they state what their focusing efficiency is the present paper.

There are some points that could be more clear but are not major:

1) the authors might want to consider amplifying their explanation of how the design process works. If I understand it, the distribution of the actual phase across lower surface has to agree with Equation (2), and nonlinear least squares is used to solve for beta and phi to achieve that. I thought this could be stated more succinctly.

2) A stylistic point: on page 3, last sentence of top paragraph "Therefore, creating acoustic metasurface without cumbersome structures would be of both fundamental and practical significance, but yet remains a critical barrier towards their realizations" is awkwardly written. Maybe, "but the complexity of the current structures remain a critical barrier." Seemed like the writing could be economized here.

3) On page 5, equation (1), the $A_p(r)$ introduction is abrupt and you do not differentiate it from p on the previous page. It might be clearer to state $A = \beta \exp(i\phi)$. In $p(r)$ you are just denoting a cylindrical wave. what is $A_p(r)$.

4) It is common for the Hankel function to have its superscript in parentheses.

5) On Page 6: While a reader could look up the definition of a Fresnel number, it would be more convenient if you state explicitly what it is. On the next page can you define what BW represents

for clarity?

6) A stylistic point: On Page 9, sentence "For the same reason, it can be seen some unwanted sidelobes". Perhaps, "For the same reason, some unwanted sidelobes can be seen distributed around the vertical boundaries". This might be stylistic but just sounded better to me.

7) in the supplementary materials:

- supplementary Figure 3 caption. "The slit thickness, spacing, and number of the acoustic metasurface are"

do you mean the number of slits?

- 2nd expression in the sentence after Equation A-8: not sure what this second expression is adding.

$\exp(i \cdot \alpha \cdot x)$ cannot equal $(1/w) \cdot \int_{-w/2}^{w/2} \exp(i \cdot \alpha \cdot x) \cdot dx$. (since x dependence has to integrate out).

- 2nd expression in Equation A.9, are you sure the delta function is needed there? isn't it already taken care of in A.3?

Reviewer #2:

Remarks to the Author:

The authors propose a metasurface design based on the use of sub-wavelength slits on a thin plate, where also the thickness of the plate and the inter-slit distances are highly sub-wavelength. They use model an acoustic version of the "double-slit" to demonstrate the existence of interactions between adjacent slits and design two lenses, one with a focus at 3λ and one with a focus in the near field.

The idea is simple and powerful, and therefore should be considered for publication, but there are some significant points that need to be addressed. Some of them are summarised below. Please note that I am not commenting on the model, due to time constraints.

1. Other authors have been exploring the use of simple, non-labyrinthine "holes" in surfaces to obtain phase sculpting of acoustic wave fronts (e.g. <https://aip.scitation.org/doi/full/10.1063/1.4972407> <https://www.nature.com/articles/srep19519>) Please comment on these works, highlighting their differences with yours.

2. While potentially novel in acoustics, the use of sub-wavelength patterned structures is well known in optics and fluid dynamics. It is well known that in these cases there are interactions between the holes/slits: how did the authors consider these studies?

3. The authors state that "the geometrical complexity and low efficiency of Fresnel lenses impair their applications in practice. Furthermore, the focusing capability of these bulk lenses deteriorates when their dimensions are reduced toward wavelength-sized scale." With this in mind, can the authors still label their work as Fresnel lenses? Maybe a change in the title/introduction/abstract would be desirable?

4. On labyrinthine metasurfaces, the authors specify that "given their complex geometries and narrow channels, they inevitably cause fabrication difficulties and suffer from large viscosity even when their dimensions become smaller". The difficulty of manufacturing labyrinthine solutions is a

fact, but the “large viscosity” is purely an optimisation issue: not all the authors were concerned with how much energy was transmitted. Some, however, exploited this issue in a favourable way, like [10] and [44]. Please comment on this.

5. Fig. 2 describes the case of a double slit, which is a well-known experiment in optics/water waves. While the authors should stress that this simple experiment has not been attempted before with their parameters, results should therefore be expected. Please highlight the differences.

6. Please clarify where you evaluated the pressure in figures 2b and 2c. Also, presenting the change in pressure/phase would be more helpful if you reported the input pressure and the reference pressure (for calculating the difference).

7. One important aspect of labyrinthine designs is the potential to achieve high transmission, while still achieving the desired phase distribution. Would the authors clarify how much of the incoming energy reached the other side of their plate?

8. Does the transmission depend on the thickness of the plate? (in the range of thicknesses explored, the authors say there is no dependence...but then explain some of the experimental results with the presence of boundary effects)

9. It is not clear what can be seen in Fig 2e, as this picture is only mentioned in the text. How does the fit compare with the analytical theory proposed by the authors? What is the meaning of the constants in the fit? (one would expect no effect with $t=0$)

10. Please comment 3d considering the Rayleigh limit for diffraction (i.e. how small can be the spot of a lens given the presence of diffraction). This should reinforce some of the conclusions towards sub-wavelength imaging.

11. Please clarify at which distance you wanted the focus to be in Fig. 4 and zoom on that region for showing the differences between theory and experiments.

12. Methods: which frequency was used for the experiments? How did the authors ensure that they had an input plane wave?

13. General: there are some sentences that need to be checked, as they may have been “lost in translation” and sound weird in English.

14. Please highlight clearly the differences with references 28-31. This will indeed be a chance to highlight the novelty of this work..or to show that you are applying to acoustics (for the first time?) ideas well-accepted in other sciences.

We thank the referees for their constructive comments and generous suggestions on the manuscript, which led to an overall strengthening of the work. We have studied the comments carefully and made corresponding revisions which are highlighted in red in the revised manuscript. For clarity, the main changes made in the main text and in the Supplementary Information are summarized in a separate file “List of Changes for NCOMMS-18-11459A-Z”.

Reviewer #1 (Remarks to the Author):

[1] *I think it's an interesting paper, and suitable for publication in Nature Communications. Technically it is well argued, the methods are clear, and there is enough information supplied that one could reproduce results. The design method is definitely semi-analytical in that you need a fairly sophisticated code to extract the complex amplitude of the cylindrical waves (beta and phi). But with this info, it allows a design approach that is otherwise harder to implement by brute force.*

My only concern is whether the significance of the results are a bit over-stated (from a practical standpoint). Specifically, the focus that is observed in Figure 3b is only a doubling in amplitude from the main lobe to the first side-lobe. So that's not a strong focus. Given that the slits are so deep-sub-wavelength in width and thickness, it is a bit surprising that the wave fields can conspire to produce even that much of a focus. In other words, features so small generally do not create large scattering unless resonance is in play. But the reasoning that significant phase modification is coming from “circulation” (repeated funneling and multiple scattering), explains the effect well.

Response: We gratefully thank the referee for the expert review and insightful comments.

[2] *But it is unfortunate that a computational case is not presented with a wider overall aperture to test the limits of the approach.*

Response: Thank you for the reminder. Indeed, the overall aperture (D) of the acoustic metasurface is limited for a given focal length (f) in our work. According to the Abbe criterion, the diffraction-limited spot size is $d = \lambda / 2NA$, where $NA = D / 2f$ is the numerical aperture of the acoustic metasurface. For the far-field focusing, the focal spot size should be larger than the diffraction limit, which is $\lambda/2$, due to the loss of evanescent

waves. Accordingly, NA should be smaller than unity and **the theoretical largest overall aperture is $D = 2f$** . It is also worth noting that it is difficult to reach this limit as the phase match became progressively difficult for the increasing NA .

The aperture limitation also exists in the near-field focusing. This is because the contributions of sub-diffraction focusing mainly come from the evanescent waves. However, due to their exponentially decaying nature, the contributions of evanescent waves emitting from the outer slits rapidly decrease when they move away from the lens center. Meanwhile, as the metasurface lens is not operating in resonant mode, the surface acoustic evanescent waves can't be efficiently excited, and therefore the coupling of outer slits through funneling and scattering is also negligible. Overall, a limited overall aperture is sufficient for near-field focusing.

As reminded by the referee, we have added this part in the revised manuscript. Please refer to: **“Moreover, it is necessary to point out that the overall aperture of the far-field metasurface lens is limited, that is $(N - 1)s \leq 2f$. Also, a limited overall aperture is sufficient for near-field focusing as the evanescent waves are exponentially decaying and the contributions from outer slits decrease rapidly when they move away from the lens center.”** in Page 11, Lines 1-5.

[3] *Another general comment has to do with the applicability of the term “Fresnel lens”. The “Fresnel” aspect is a bit of a misuse of the term in a sense that a Fresnel zone concept is not at play. But there are slits in a plate making an aperture so it is a bit related to Fresnel lenses. Also, the authors mention low focusing efficiency of Fresnel lenses, but I don't believe they state what their focusing efficiency is the present paper.*

Response: We thank the referee for these insightful comment and useful advice. We totally agree that the critical importance of our work is not the focusing efficiency, and the term of “Fresnel lens” is a bit of a misuse. The focus of this work is to clarify the underlying physics for acoustic wave manipulations by considering the full wave dynamics between unit elements of the acoustic metasurface in deep-subwavelength scale and presents a semi-analytical microscopic approach to simply optimize the designed lens, instead of intensive trials on numerical simulations.

To address the referee's concern and better reflect the basic idea of our manuscript, we have revised the term “Fresnel lens” throughout the revised manuscript, including the title,

abstract, and main text. To be consistent with this change, the characteristic parameter of acoustic lens FN (Fresnel number) is replaced by NA (numerical aperture).

[4] *There are some points that could be more clear but are not major:*

1) *the authors might want to consider amplifying their explanation of how the design process works. If I understand it, the distribution of the actual phase across lower surface has to agree with Equation (2), and nonlinear least squares is used to solve for beta and phi to achieve that. I thought this could be stated more succinctly.*

Response: We thank the referee for this constructive advice. Following this suggestion, we specify the design process in the revised manuscript as “**Given the assigned structural parameters and the established microscopic model, the slit widths in the acoustic metasurface were numerically optimized with nonlinear least squares fitting (see Methods), which minimizes the differences between the actual phase across the exit surface and that from Eq. (2). The optimized slit widths are listed in Supplementary Table 1**” in Page 7, Lines 13-17.

2) *A stylistic point: on page 3, last sentence of top paragraph “Therefore, creating acoustic metasurface without cumbersome structures would be of both fundamental and practical significance, but yet remains a critical barrier towards their realizations” is awkwardly written. Maybe, “but the complexity of the current structures remain a critical barrier.” Seemed like the writing could be economized here.*

Response: Many thanks for the suggestion. According to this advice, we have rephrased the sentence as “**Therefore, creating acoustic metasurfaces without cumbersome structures would be of both fundamental and practical significance, but the complexity of the current structures remains a critical barrier.**” in Page 3, Lines 5-7. To avoid similar problems and facilitate the readability, we have asked a native English speaker to polish the revised manuscript.

3) *On page 5, equation (1), the $A_p(r)$ introduction is abrupt and you do not differentiate it from p on the previous page. It might be clearer to state $A = \beta \exp(i\phi)$. In $p(r)$ you are just denoting a cylindrical wave. what is $A_p(r)$.*

Response: We thank the referee for pointing out this problem. $A_p(r)$ denotes the

distribution of the scattered CW spreading out from a subwavelength slit. As suggested by the referee, and for clarity, we have changed $Ap(r)$ to $A(r) = \beta e^{i\phi} H_0^{(2)}(k_0 r)$ and introduced it before Eq. (1). Please refer to “For a scattered CW spreading out from a subwavelength slit, its distribution can be described as $A(r) = \beta e^{i\phi} H_0^{(2)}(k_0 r)$ ³⁹, where β and ϕ are the scattering coefficient and abrupt phase jump of the CW; r is the distance from the slit center and $H_0^{(2)}$ is the zeroth-order Hankel function of the second kind. Accordingly, the pressure field of the excited CW can be written as” from Page 5, Line 31 to Page 6, Line 3. The other parts are correspondingly revised with this change.

4) *It is common for the Hankel function to have its superscript in parentheses.*

Response: According to the referee’s advice, we have corrected the Hankel function as $H_0^{(2)}$.

5) *On Page 6: While a reader could look up the definition of a Fresnel number, it would be more convenient if you state explicitly what it is. On the next page can you define what BW represents for clarity?*

Response: We thank the referee for the valuable advice. According to this suggestion, we have explicitly stated the corresponding terms in the revised manuscript. We regret the confusion arising from the abrupt appearance of the term BW . In fact, it denotes the size of the diffraction-limited focal spot. As suggested by the referee, we have clarified the term as “the focal spot size BW is calculated to be 1.02λ with $BW = 0.51\lambda / NA$ ” in Page 7, Lines 27-28.

It needs to be pointed out that FN has been replaced by NA in the revised manuscript as we have corrected the misuse of “Fresnel lens” (see the **Response** to [3]).

6) *A stylistic point: On Page 9, sentence “For the same reason, it can be seen some unwanted sidelobes”. Perhaps, “For the same reason, some unwanted sidelobes can be seen distributed around the vertical boundaries”. This might be stylistic but just sounded better to me.*

Response: We appreciate the referee’s help. According to this suggestion, we have revised the sentence as “For the same reason, some unwanted sidelobes can be seen distributed

around the vertical boundaries (white dashed circles).” in Page 9, Lines 27-29.

7) In the supplementary materials:

- Supplementary Figure 3 caption. “The slit thickness, spacing, and number of the acoustic metasurface are” do you mean the number of slits?

Response: Yes, we thank the referee for pointing out this problem. In the Supplementary Information, we have revised it to “The thickness, spacing, and slit number in the acoustic metasurface are 0.1λ , 0.15λ and 19, respectively.”

- 2nd expression in the sentence after Equation A-8: not sure what this second expression is adding. $\exp(i\alpha x)$ cannot equal $(1/w) \int_{-w/2}^{w/2} \exp(i\alpha x) dx$. (since x dependence has to integrate out).

Response: We regret our use of this undefined expression. In fact, this is an approximated evaluation of $e^{i\alpha x}$. Although the slits are deep-subwavelength compared to the incident wave, they cannot be simply considered as points for all waves, especially the evanescent waves with large tangential wave-vectors. As the integration is performed for all wave components over $[-\infty, \infty]$, we therefore generalize the approximation of $e^{i\alpha x}$ by averaging it on the whole slit width $[-w/2, w/2]$, which is $e^{i\alpha x} \approx \frac{1}{w} \int_{-w/2}^{w/2} \exp(i\alpha x) dx$.

As reminded by the referee, we have added a comment in Supplementary Information to clarify this expression as “Note that although the slits are deep-subwavelength compared to the incident wave, they cannot be simply considered as points for all waves, especially the evanescent waves with large tangential wave-vectors. As the integration is performed for all wave components over $[-\infty, \infty]$, here we generalize the evaluation of $e^{i\alpha x}$ by averaging it on the whole slit width, to integrate out the x dependence. Thus, we have

$e^{i\alpha x} \approx \frac{1}{w} \int_{-w/2}^{w/2} \exp(i\alpha x) dx = \frac{2 \sin(w\alpha/2)}{\alpha w}$.” in Page 6, Lines 17-20.

- 2nd expression in Equation A.9, are you sure the delta function is needed there? isn't it already taken care of in A.3?

Response: We thank the referee for pointing out this mistake. We have fixed it in the Supplementary Information.

Reviewer #2 (Remarks to the Author):

The authors propose a metasurface design based on the use of sub-wavelength slits on a thin plate, where also the thickness of the plate and the inter-slit distances are highly sub-wavelength. They use model an acoustic version of the “double-slit” to demonstrate the existence of interactions between adjacent slits and design two lenses, one with a focus at 3λ and one with a focus in the near field.

The idea is simple and powerful, and therefore should be considered for publication, but there are some significant points that need to be addressed. Some of them are summarised below. Please note that I am not commenting on the model, due to time constraints.

Response: We thank the referee for the positive remarks and valuable advice. We have made every effort to revise and improve the manuscript.

1. Other authors have been exploring the use of simple, non-labyrinthine “holes” in surfaces to obtain phase sculpting of acoustic wave fronts (e.g. <https://aip.scitation.org/doi/full/10.1063/1.4972407>, <https://www.nature.com/articles/srep19519>). Please comment on these works, highlighting their differences with yours.

Response: We thank the referee for recommending these related works. For convenience, we reference them as APL work (<https://aip.scitation.org/doi/full/10.1063/1.4972407>) and SciRep work (<https://www.nature.com/articles/srep19519>), respectively.

In APL work, the wavefront used to generate an acoustic tractor beam was sculpted by controlling the single-pass phase delay through each waveguide (straight or coiled tube). This single-pass phase delay can be effectively tuned by adjusting the effective path length, as proposed in the APL work with three different solutions. However, in our work, such kind of phase delay is trivial because of the deep-subwavelength thickness. Instead, the phase is effectively modulated through the strong couplings between the slits which are deep-subwavelength spaced. Thus, the origin of phase modulation is different between the APL work and our work.

In SciRep work, a sub-wavelength multi-resonant scatterer was proposed for perfect acoustic absorption by controlling the interplay between the energy leakage of the resonances into the waveguide and the inherent losses. Once the leakage and the inherent loss were balanced, a critical coupling condition was fulfilled and the resulting destructive

interference between the transmitted and the internal fields produced zero reflection at the exit of the waveguide. Accordingly, maximum acoustic absorption was achieved at the resonance frequency. Furthermore, through the balance control of several resonances, multiple peaks can be produced, and broadband perfect absorption can be realized upon their overlapping. There is similarity on the adjustment of acoustic response of single waveguide between this work and ours (e.g., unit length in SciRep work and unit width in our work). However, in our work, the wave dynamic at each slit is not only controlled by its own acoustic response but also by the other slits through strong couplings because of the deep-subwavelength spacing. The critical importance of our work is to reveal the whole wave dynamics between the slits, based on which a comprehensive design rule is established to optimize the acoustic metasurface for wave manipulations, either in far- or near-field.

For the reference, we have cited these two works in the revised manuscript. Notably, these recommendations are very useful, shedding light on some follow-up studies to extend the wave manipulations through revisiting a number of existing approaches.

2. While potentially novel in acoustics, the use of sub-wavelength patterned structures is well known in optics and fluid dynamics. It is well known that in these cases there are interactions between the holes/slits: how did the authors consider these studies?

Response: We agree with the referee's insightful comment. Indeed, the sub-wavelength patterned structures have been widely exploited in optics and fluid dynamics, and can be divided into two main categories: periodic structures and aperiodic structures. For the periodic structures, because of the subwavelength nature of the unit elements, they are often believed to be widely governed by near-field interactions and effective medium theory are normally used to study their macroscopic properties, yielding anomalous phenomena such as negative refraction, extraordinary transmission, extreme refractive index, cloaking, among others. However, the periodic structures are generally frequency selective, lacking the capability to form spatially varying responses to manipulate wavefronts into shapes at will. Aperiodic structures have also been explored for wave manipulations, including focusing such as the work in Refs [41] (Nano Lett. 9, 235-238, 2009) and [42] (JASA 130(5), 2789-2796, 2011). In [41], the planar metalens for optical focusing was designed based on the phase delay of individual slits, which links to the slit

width through the dispersion relationship of metal-insulator-metal structure. However, both in simulation and experiment, the actual focal behavior was significantly deviated from the prediction. This large discrepancy mainly arises from the insufficient considerations of the wave dynamics occurring in the structure. In [42], a simulation-based computational model was proposed for acoustic focusing by optimizing an aperiodic ring structure. This approach is highly dimensional and computation intensive as the structure needs to be divided into a large number of elements, i.e., 127 in the case study. More importantly, it is very difficult to find the global solution for such an optimization problem with great inherent complexity. For this reason, both the focal length (2.5 mm) and spot size (3.5 mm) of the optimized lens vastly disagree with their designations, which are 6.7 mm and 1.7 mm, respectively. Similarly, the underlying physics is not clear and the design process is by trial.

As reminded by the referee, we have added a comment in the revised manuscript to consider this issue. Please refer to “It is also worth noting that the periodic sub-wavelength patterned structures have been widely exploited in optics and fluid dynamics, and effective medium theory³⁴ is generally employed to study their macroscopic properties as they are often believed to be governed by near-field interactions. However, this theory also cannot be applied to our design.” in Page 3, Lines 25-28.

Also, we have revised the sentences as “Compared with the previous design rules for the slit structure, such as using single-pass phase retardation⁴¹ or simulation-based numerical optimization⁴², the semi-analytical approach presented here worked in a simple, straightforward and accurate manner, and achieved a much better performance on the prediction of the focusing behavior.” in Page 10, Lines 25-28.

3. The authors state that “the geometrical complexity and low efficiency of Fresnel lenses impair their applications in practice. Furthermore, the focusing capability of these bulk lenses deteriorates when their dimensions are reduced toward wavelength-sized scale.” With this in mind, can the authors still label their work as Fresnel lenses? Maybe a change in the title/introduction/abstract would be desirable?

Response: We thank the referee for the insightful comment and valuable advice. Following this suggestion, we have revised the term “Fresnel lens” throughout the revised manuscript, including the title/introduction/abstract. Meanwhile, to be consistent with this change, the

characteristic parameter FN (Fresnel Number) is replaced by NA (Numerical Aperture).

4. *On labyrinthine metasurfaces, the authors specify that “given their complex geometries and narrow channels, they inevitably cause fabrication difficulties and suffer from large viscosity even when their dimensions become smaller”. The difficulty of manufacturing labyrinthine solutions is a fact, but the “large viscosity” is purely an optimisation issue: not all the authors were concerned with how much energy was transmitted. Some, however, exploited this issue in a favourable way, like [10] and [44]. Please comment on this.*

Response: We thank the referee for pointing out this inaccurate statement. As commented by the referee, the presence of thermal and viscous losses may affect the wave transmission, but this problem can be readily mitigated by optimizing the coiling-up-space structures to achieve high transmission. On the other hand, there also has been a growing interest in exploring new physics by embracing the losses in acoustic metamaterials, such as independent phase and amplitude control, asymmetric transmission, among others. To correct this statement, we have rewritten the sentence as “**On the other hand, given their complex geometries and narrow channels, they inevitably cause fabrication difficulties, particularly when their dimensions become smaller.**” in Page 3, Lines 2-4.

5. *Fig. 2 describes the case of a double slit, which is a well-known experiment in optics/water waves. While the authors should stress that this simple experiment has not been attempted before with their parameters, results should therefore be expected. Please highlight the differences.*

Response: We thank the referee for this valuable advice. As pointed out by the referee, the double-slit experiment is well-known in optics and water waves, and has become a classic experiment in expressing central puzzles. While conventional double-slit experiments are mainly focusing on the investigation of wave interference patterns, the slits are generally separated with a considerable distance. Accordingly, the coupling between the slits is trivial and they can be considered as independent radiating point sources. However, this is not the case when the slit separation is in deep-subwavelength scale where the coupling becomes prominent. Under this circumstance, the diffractive acoustics of one slit will be markedly influenced by the other one as well as their inter-distance through the coupling effect.

Following the referee's suggestion, we have added a sentence to stress this point as "While the double-slit experiment is well-known in optics and water waves, it has not been attempted yet when the slits are separated with a deep-subwavelength distance, with which the coupling effect becomes prominent." in Page 4, Lines 18-20.

6. Please clarify where you evaluated the pressure in figures 2b and 2c. Also, presenting the change in pressure/phase would be more helpful if you reported the input pressure and the reference pressure (for calculating the difference).

Response: We thank the referee for this valuable advice. The pressure in Figs. 2b and 2c is evaluated at the exit of the reference slit (left one in Fig. 2a). The pressure of the input plane wave is 1 Pa, and the reference pressure for the single slit is numerically computed to be $0.611-0.125i$, whose amplitude and phase are 0.624 Pa and -0.2 radians, respectively. As suggested by the referee, we have added these details in the revised manuscript for clarity. Please refer to "We define this effect as $\Delta p = p' - p_0$ and $\Delta\phi = \phi' - \phi_0$, where p' , ϕ' are the amplitude and phase of the pressure field extracted at the exit of the reference slit in doublet configuration and p_0 , ϕ_0 are those from the single reference slit. For simplicity, the pressure of the incident plane wave was normalized to 1 Pa. The direct transmission of the reference slit was numerically computed to be $0.611-0.125i$ Pa, that is $p_0 = 0.624$ Pa and $\phi_0 = -0.2$ radians." in Page 4, Lines 25-30.

7. One important aspect of labyrinthine designs is the potential to achieve high transmission, while still achieving the desired phase distribution. Would the authors clarify how much of the incoming energy reached the other side of their plate?

Response: According to the referee's suggestion, we have numerically computed the incident and transmitted energy using the domain integration in COMSOL. The intensity is integrated over the transmission domain ($-2\lambda \leq x \leq 2\lambda$, $0 \leq z \leq 8\lambda$), and the transmitted and incident energy are $17.3 \mu W$ and $37.8 \mu W$, respectively. As a result, 45.8% of the incoming energy can reach the other side.

As reminded by the referee, we have added it in the revised manuscript. Please refer to "Meanwhile, the incident and transmitted energy were numerically computed by integrating the intensity over the transmission domain ($-2\lambda \leq x \leq 2\lambda$, $0 \leq z \leq 8\lambda$), which

were 37.8 μW and 17.3 μW , respectively. Thus, a transmittance of 45.8% can be obtained for the designed metasurface lens.” in Page 7, Lines 28-31.

8. *Does the transmission depend on the thickness of the plate? (in the range of thicknesses explored, the authors say there is no dependence...but then explain some of the experimental results with the presence of boundary effects)*

Response: We are sorry for the ambiguity. In fact, the thickness will affect the transmission, which can be seen in the insets of Fig. 3d, in which the peak amplitude varies with the thickness. However, the deep-subwavelength thickness has small influence on the focusing behavior. This can be attributed to following aspects: (1) only phase is accounted for the far-field focusing; (2) the phase modulation mainly comes from the coupling effects which are tuned by the slit widths; (3) the single-pass phase delay through each slit is trivial when the thickness is deep-subwavelength in scale. For this reason, although the amplitude changes, the focal length and spot size remain nearly the same for different thicknesses (in the range of $[0.01\lambda, 0.1\lambda]$), as shown in Fig. 3d. For clarity, we have revised the sentences as “Although the transmission changed with the thickness, as shown in the insets, the *FWHM* and focal length were nearly the same, whereas the thickness varied in the range of $[0.01\lambda, 0.1\lambda]$. This can be expected as only phase is taken into account in the far-field sound focusing.” in Page 8, Lines 16-19.

Regarding the boundary effect, it comes from the diffractions of the incident wave from two outer boundaries of the fabricated lens because of its finite width, which is infinite in both numerical simulation and theoretical calculation. According to the numerical studies, the slight amplitude rise of the sidelobes in experiment (Fig. 3b) testifies the presence of such effect. To clarify this point, we have added a comment as “Also, it was affected by the diffractions of the incident wave from the outer edges of the metasurface lens because of its finite width. The presence of such diffractions was testified by the slight amplitude rise of the sidelobes in experiment in Fig. 3b.” in Page 8, Lines 5-8.

9. *It is not clear what can be seen in Fig 2e, as this picture is only mentioned in the text. How does the fit compare with the analytical theory proposed by the authors? What is the meaning of the constants in the fit? (one would expect no effect with $t=0$)*

Response: We thank the referee for pointing out this problem. Figure 2e shows the dependence of the complex amplitude of the scattered cylindrical waves (CWs) on slit width. With this knowledge, the wave dynamics in the metasurface can be calculated and then tuned by adjusting the slit widths. In theory, it is viable to obtain this knowledge by repeatedly calculating the complex amplitude for possible slit width by using Eq. (1) and generate the large database that serves as the look-up library. However, this process will be tedious. Here we gain the knowledge in an economical way by fitting the complex amplitudes extracted from 40 slit widths, ranging from 0.005λ ($t = 0.5$) to 0.2λ ($t = 20$) with a step of 0.005λ ($t = 0.5$). Notably, as pointed out by the referee, the fitted curve doesn't coincide well when the slit width approaches zero. Actually, a deviation can be readily seen at $t = 0.5$. Nevertheless, we think this will not be a big problem as a lower bound is generally set on the slit width in the metasurface optimization to facilitate the fabrication in reality.

In the revised manuscript, we clarify this problem as “To link the characteristic parameters of the CW to the slit width, β and ϕ are fitted as a polynomial function with the least-mean-square method and shown in Fig. 2e. Note that the slit width is scaled in the fitting, i.e., $t = w / 0.01\lambda$. It is also worth noting that the fit doesn't coincide well when the slit width approaches zero, and a small deviation can be seen at $t = 0.5$. Nevertheless, we think this will not be a big problem as a lower bound is generally set on the slit width in the metasurface design to facilitate the fabrication (see Methods).” in Page 6, Lines 15-20.

10. *Please comment 3d considering the Rayleigh limit for diffraction (i.e. how small can be the spot of a lens given the presence of diffraction). This should reinforce some of the conclusions towards sub-wavelength imaging.*

Response: We thank the referee for the valuable suggestion. According to the Rayleigh criterion, the far-field lens would give a diffraction limit of 0.61λ , whereas the *FWHM* of the focal spot in simulations was found to be approximately 1.0λ . For a given focal length, the focal spot size depends on the overall aperture of the metasurface: the larger the aperture, the higher the *NA*, and the tighter the focus. However, the focal spot of far-field lens is restricted by the diffraction limit due to the loss of evanescent waves. To break this limit and achieve sub-diffraction focusing, the evanescent waves should be incorporated, which are bound to the near field.

As suggested by the referee, we have added this comment in the revised manuscript. Please refer to “Also, for a given focal length, the focal spot size depends on the overall aperture of the metasurface lens: the larger the aperture, the higher the NA , and the tighter the focus. According to the Rayleigh criterion, the far-field lens would give a diffraction limit of 0.61λ , whereas the $FWHM$ of the focal spot in simulations was found to be approximately 1.0λ . It is worth noting that although the focal spot of the far-field lens can be further narrowed by increasing the overall aperture, it will be restricted eventually by the diffraction limit due to the loss of evanescent waves. To break this limit and achieve sub-diffraction focusing, the evanescent waves, which are bound to the near field, should be incorporated.” in Page 8, Lines 21-28.

11. Please clarify at which distance you wanted the focus to be in Fig. 4 and zoom on that region for showing the differences between theory and experiments.

Response: We thank the referee for pointing out this issue. According to this suggestion, we have added a white dashed line in the middle panel of Fig. 4a (simulation plot) to indicate the focal depth. For clarity, the differences between the theory and experiment are also circled out in the right panel of Fig. 4a (experiment plot).

Fig. 4a

12. Methods: which frequency was used for the experiments? How did the authors ensure that they had an input plane wave?

Response: In the experimental measurements, the operating frequency was set to 3.43 kHz for the far-field lens ($\lambda = 100$ mm) and 1.15 kHz for the near-field plate ($\lambda = 300$ mm), respectively. Regarding the input plane wave, we have investigated the acoustic field generated by the planar acoustic speaker. Several measurements were performed along the

45° diagonal path in front of the speaker on which the microphone slide. The measured amplitude and phase of the acoustic signals acquired at selective frequencies are plotted in Fig. S1. From Fig. S1, we can find that the amplitude changes slightly when the distance increases while the phase decreases linearly. As a result, it can be concluded that the acoustic field is closely plane-wave. As pointed out by the referee, we have added Fig. S1 in the Supplementary Information (Supplementary Fig. 4).

Fig. S1 | (a) The amplitude and (b) phase of the acoustic signals generated by the plane wave speaker at selected frequencies: 1942Hz (circles), 3294Hz (squares), 4129Hz (diamonds) and 5185Hz (triangles).

13. *General: there are some sentences that need to be checked, as they may have been “lost in translation” and sound weird in English.*

Response: We are sorry for the inconvenience and confusions caused by the linguistic problems. According to your suggestion, we have asked a native English speaker to proofread the revised manuscript.

14. *Please highlight clearly the differences with references 28-31. This will indeed be a chance to highlight the novelty of this work or to show that you are applying to acoustics (for the first time?) ideas well-accepted in other sciences.*

Response: We thank the referee for the constructive advice. Following this suggestion, we have rewritten this part to highlight the differences with references 28-31. Please refer to “In general, the acoustic response of each slit is individually tuned by adjusting its size^{27,28}, localized resonance^{29,30}, or filling material^{9,31}. The wave couplings between the slits can be neglected if they are deep enough and therefore the slits are considered to be independent.

However, this is not the case when the thickness of the plate, and thus depth of the slits, is much smaller than the wavelength. For this reason, the simple design rule based on the properties of individual slits would result in a significant discrepancy in focusing. Such discrepancy becomes more severe when spacing of the slits is also much smaller than the wavelength. Several studies have investigated the causes of the focal shift effect, but they are mainly focusing on the influences of the structural parameters, such as slit number, focal length and lens width^{32,33}. Indeed, these parameters will affect the focusing behavior, but the underlying physics on how they change the wave dynamics and modify the field distribution remain to be clarified. Meanwhile, the simple design rule faces another difficult challenge that the tuning capability for a single slit becomes very poor if both its width and thickness are deeply subwavelength.” in Page 3, Lines 12-25.

Reviewers' Comments:

Reviewer #1:

Remarks to the Author:

All points raised during reviews have been acceptably addressed in the revision.

Reviewer #2:

Remarks to the Author:

I would like to thank the authors for considering all my comments. I find the manuscript greatly improved and it is now ready for publication.

Just a comment on the speaker.

The font used for Fig S1 (now supplemental figure 4) is very small and hard to read in print. Once properly zoomed, this picture shows that:

a) the amplitude does not change along the 45deg line in the range 1.35-1.6 m. With a spherical source, one would expect a decrease of about 1.5 dB...which does not appear.

b) the phase changes differently depending on the frequency chosen.

In my experience, this may indicate the presence of reflections due to the objects surrounding the area of measurement and/or the presence of lobes at some frequencies.

These could explain the difference between theory and experiment observed by the authors (i.e. a diffused field to the sides of the lens).

My suggestion, in future studies, would be to scan the emission of the selected speaker along a line perpendicular to the direction of propagation to actually characterise its shape.

Reviewer #1 (Remarks to the Author):

All points raised during reviews have been acceptably addressed in the revision.

Response: Thank you very much.

Reviewer #2 (Remarks to the Author):

I would like to thank the authors for considering all my comments. I find the manuscript greatly improved and it is now ready for publication.

Response: We thank the referee for the positive remark and the valuable suggestions that greatly improve our work.

Just a comment on the speaker:

1. *The font used for Fig S1 (now supplemental figure 4) is very small and hard to read in print.*

Response: We thank the reviewer for pointing out this problem. As reminded by the reviewer, we have zoomed the font size in Supplementary Figure 4.

Supplementary Figure 4

2. *Once properly zoomed, this picture shows that:*

a) the amplitude does not change along the 45deg line in the range 1.35-1.6 m. With a spherical source, one would expect a decrease of about 1.5 dB...which does not appear.

b) the phase changes differently depending on the frequency chosen.

In my experience, this may indicate the presence of reflections due to the objects surrounding the area of measurement and/or the presence of lobes at some frequencies.

These could explain the difference between theory and experiment observed by the authors (i.e. a diffused field to the sides of the lens).

Response: We are sorry for the confusion on this issue due to some missing details. The source used in the experiments is "Panphonics Sound Shower" speaker, according to the manufacturer, which is able to generate plane wave in the far field. To further confirm that, we measured the free-space acoustic field emitting from the speaker at different frequencies in an anechoic room. Herein, the measurements were performed along a 45deg line in front of the speaker, and the distance of the measurement locations was recorded with respect to a reference point on the frame diagonal. The measured amplitude and phase are shown in the Supplementary Figure 4, both of which indicate that the acoustic field is closely plane-wave. **For example, (a) the amplitude shows insignificant change at different distances, instead of 1.5dB decrease that is expected with a spherical source; (b) the phase changes linearly with respect to distance while the slope of phase change depends on the frequency: the higher the frequency, the larger the slope.** For a quantitative confirmation, we further fit the measured phase using linear least squares, and the parameters are listed in table S1. It can be seen that the fitted slopes agree well with their theoretical counterparts that are calculated from $d\phi = -2\pi f \cos(45^\circ) / c_{\text{air}}$, where $d\phi$ is the slope, f is the frequency and $c_{\text{air}} = 343\text{m/s}$ is the sound speed in air. Thus, we deem that the acoustic field is directive and closely plane-wave within the operating frequency range.

To clarify this issue, we have added table S1 to the Supplementary Information (Supplementary Table 3) and a comment in the legend of Supplementary Figure 4. Please refer to **“The measurements were performed along the 45° diagonal path in front of the speaker on which the microphone slides. The distance of the measurement locations was recorded with respect to a reference point on the frame diagonal. The measured phase was fitted using linear least squares, and the parameters are listed in Supplementary table 3.”**

Meanwhile, as pointed out by the reviewer, we cannot exclude the presence of small lobes at large off-axis angles with respect to the axis normal to the speaker, which may cause the difference between theory and experiment. Also, we cannot exclude the presence of some unwanted reflections from surrounding objects, but we think such reflections

would not have a significant effect on the results as the measurements were carried out in the anechoic room.

Table S1: The linear least square fitting parameters of the measured phases

Frequency (Hz)	Theoretical phase slope (rad m ⁻¹)	Experimental phase slope (rad m ⁻¹)	R ² value
1942	-25.15	-24.47	0.9954
3294	-42.67	-43.13	0.9981
4129	-53.48	-56.02	0.9986
5185	-67.16	-67.06	0.9969

3. *My suggestion, in future studies, would be to scan the emission of the selected speaker along a line perpendicular to the direction of propagation to actually characterise its shape.*

Response: We greatly appreciate the reviewer's constructive suggestion on the consideration of wavefront shape upon incidence, which is of critical importance for the accurate wave manipulation. We will follow this advice and take this factor into account in future studies.